# Causal Reasoning from Meta Reinforcement Learning

## Abstract

Discovering and exploiting the causal structure in the environment is a crucial challenge for intelligent agents. Here we explore whether modern deep reinforcement learning can be used to train agents to perform causal reasoning. We adopt a meta-learning approach, where the agent learns a policy for conducting experiments via causal interventions, in order to support a subsequent task which rewards making accurate causal inferences. We also found the agent could make sophisticated counterfactual predictions, as well as learn to draw causal inferences from purely observational data. Though powerful formalisms for causal reasoning have been developed, applying them in real-world domains can be difficult because fitting to large amounts of high dimensional data often requires making idealized assumptions. Our results suggest that causal reasoning in complex settings may benefit from powerful learning-based approaches. More generally, this work may offer new strategies for structured exploration in reinforcement learning, by providing agents with the ability to perform—and interpret—experiments.

## 1 Introduction

Many machine learning algorithms are rooted in discovering patterns of correlation in data. While this has been sufficient to excel in several areas (Krizhevsky et al., 2012; Cho et al., 2014), sometimes the problems we are interested in are fundamentally causal. Answering questions such as "Does smoking cause cancer?" or "Was this person denied a job due to racial discrimination?" or "Did this marketing campaign cause sales to go up?" all require an ability to reason about causes and effects and cannot be achieved by purely associative inference. Even for problems that are not obviously causal, like image classification, it has been suggested that some failure modes emerge from lack of causal understanding. Causal reasoning may be an essential component of natural intelligence and is present in human babies, rats and even birds (Leslie, 1982; Gopnik et al., 2001; 2004; Blaisdell et al., 2006; Lagnado et al., 2013). There is a rich literature on formal approaches for defining and performing causal reasoning (Pearl, 2000; Spirtes et al., 2000; Dawid, 2007; Pearl et al., 2016).

Here we investigate whether procedures for learning and using causal structure can be produced by meta-learning. The approach of meta-learning is to learn the learning (or inference) procedure itself, directly from data. We adopt the specific method of Duan et al. (2016) and Wang et al. (2016), training a recurrent neural network (RNN) through model-free reinforcement learning. We train on a large family of tasks, each underpinned by a different causal structure.

The use of meta-learning avoids the need to manually implement explicit causal reasoning methods in an algorithm, offers advantages of scalability by amortizing computations, and allows automatic incorporation of complex prior knowledge (Andrychowicz et al., 2016; Wang et al., 2016; Finn et al., 2017). Additionally, by learning end-to-end, the algorithm has the potential to find the internal representations of causal structure best suited for the types of causal inference required.

## 2 Problem specification and approach

This work probed how an agent could learn to perform causal reasoning in three distinct settings – *observational*, *interventional*, and *counterfactual* – corresponding to different types of data available to the agent during the first phase of an episode.

In the observational setting (Experiment 1), the agent could only obtain passive observations from the environment. This type of data allows an agent to infer associations (*associative reasoning*) and, when the structure of the underlying causal model permits it, to estimate the effect that changing a variable in the environment has on another variable, namely to estimate causal effects (*cause-effect reasoning*).

In the interventional setting (Experiment 2), the agent could directly set the values of some variables in the environment. This type of data in principle allows an agent to estimate causal effects for any underlying causal model.

In the counterfactual setting (Experiment 3), the agent first had an opportunity to learn about the causal graph through interventions. At the last step of the episode, it was asked a counterfactual question of the form "What *would have* happened if a different intervention had been made in the previous time-step?".

Next we will formalize these three settings and patterns of reasoning possible in each, using the graphical model framework (Pearl, 2000; Spirtes et al., 2000; Dawid, 2007)[1], and introduce the meta-learning methods that we will use to train agents that are capable of such reasoning.

## 2.1 CAUSALITY

Causal relationships among random variables can be expressed using *causal directed acyclic graphs* (DAGs) (see Appendix). A causal DAG is a graphical model that captures both *independence* and *causal* relations. Each node $X_i$ corresponds to a random variable, and the joint distribution $p(X_1, \dots, X_N)$ is given by the product of conditional distributions of each node $X_i$ given its parent nodes $\text{pa}(X_i)$, i.e. $p(X_{1:N} \equiv X_1, \dots, X_N) = \prod_{i=1}^{N} p(X_i | \text{pa}(X_i))$.

Edges carry causal semantics: if there exists a directed path from $X_i$ to $X_j$, then $X_i$ is a *potential cause* of $X_j$. Directed paths are also called *causal paths*. The *causal effect* of $X_i$ on $X_j$ is the conditional distribution of $X_j$ given $X_i$ restricted to only causal paths.

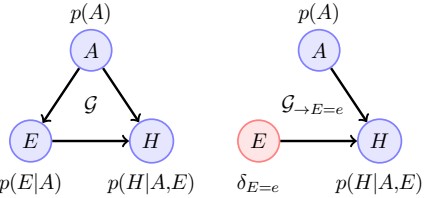

An example causal DAG $\mathcal{G}$ is given in the figure on the left, where $E$ represents hours of exercise in a week, $H$ cardiac health, and $A$ age. The causal effect of $E$ on $H$ is the conditional distribution restricted to the path $E \to H$, i.e. excluding the path $E \leftarrow A \to H$. The variable $A$ is called a *confounder*, as it confounds the causal effect with non-causal statistical influence.

Simply observing cardiac health conditioning on exercise level from $p(H|E)$ (*associative reasoning*) cannot answer if change in exercise levels cause changes in cardiac health (*cause-effect reasoning*), since there is always the possibility that correlation between the two is because of the common confounder of age.

**Cause-effect Reasoning.** The causal effect can be seen as the conditional distribution $p_{\to E=e}(H|E=e)$[2] on the graph $\mathcal{G}_{\to E=e}$ above (right), resulting from intervening on $E$ by replacing $p(E|A)$ with a delta distribution $\delta_{E=e}$ (thereby removing the link from $A$ to $E$) and leaving the remaining conditional distributions $p(H|E,A)$ and $p(A)$ unaltered. The rules of do-calculus (Pearl, 2000; Pearl et al., 2016) tell us how to compute $p_{\to E=e}(H|E=e)$ using observations from $\mathcal{G}$. In this case $p_{\to E=e}(H|E=e) = \sum_A p(H|E=e,A)p(A)$[3]. Therefore, do-calculus enables us to reason in the intervened graph $\mathcal{G}_{\to E=e}$ even if our observations are from $\mathcal{G}$. This is the scenario captured by our observational setting outlined above.

Such inferences are always possible if the confounders are observed, but in the presence of unobserved confounders, for many DAG structures the only way to compute causal effects is by collecting observations directly from $\mathcal{G}_{\to E}$, i.e. by actively intervening on the world to fix the value of the variable $E=e$ and observing the remaining variables. In our interventional setting, outlined above, the agent has access to such interventions.

---

[1]This approach typically decouples the challenges of causal induction, i.e. of inferring the structure of the causal graph from data, and that of causal reasoning on the induced graph. The formalism we describe here assumes that the structure of the causal graph is known. In our experiments however, our agents concurrently carry out causal induction.

[2]In the causality literature, this distribution would most often be indicated with $p(H|\text{do}(E=e))$. We prefer to use $p_{\to E=e}(H|E=e)$ to highlight that intervening on $E$ results in changing the original distribution $p$, by structurally altering the causal DAG.

[3]Notice that conditioning on $E=e$ would instead give $p(H|E=e) = \sum_A p(H|E=e,A)p(A|E=e)$.

**Counterfactual Reasoning.** Cause-effect reasoning can be used to correctly answer predictive questions of the type "Does exercising improve cardiac health?" by accounting for causal structure and confounding. However, it cannot answer retrospective questions about what *would have* happened. For example, given an individual $i$ who has died of a heart attack, this method would not be able to answer questions of the type "What would the cardiac health of this individual have been had they done more exercise?". This type of question requires estimating unobserved sources of noise and then reasoning about the effects of this noise under a graph conditioned on a different intervention.

## 2.2 META-LEARNING

Meta-learning refers to a broad range of approaches in which aspects of the learning algorithm itself are learned from the data. Many individual components of deep learning algorithms have been successfully meta-learned, including the optimizer (Andrychowicz et al., 2016), initial parameter settings (Finn et al., 2017), a metric space (Vinyals et al., 2016), and use of external memory (Santoro et al., 2016).

Following the approach of (Duan et al., 2016; Wang et al., 2016), we parameterize the entire learning algorithm as a recurrent neural network (RNN), and we train the weights of the RNN with model-free reinforcement learning (RL). The RNN is trained on a broad distribution of problems which each require learning. When trained in this way, the RNN is able to implement a learning algorithm capable of efficiently solving novel learning problems in or near the training distribution.

Learning the weights of the RNN by model-free RL can be thought of as the "outer loop" of learning. The outer loop shapes the weights of the RNN into an "inner loop" learning algorithm. This inner loop algorithm plays out in the activation dynamics of the RNN and can continue learning even when the weights of the network are frozen. The inner loop algorithm can also have very different properties from the outer loop algorithm used to train it. For example, in previous work this approach was used to negotiate the exploration-exploitation tradeoff in multi-armed bandits (Duan et al., 2016) and learn algorithms which dynamically adjust their own learning rates (Wang et al., 2016; 2018). In the present work we explore the possibility of obtaining a causally-aware inner-loop learning algorithm. See the Appendix for a more formal approach to meta-learning.

## 3 TASK SETUP AND AGENT ARCHITECTURE

In the experiments, in each episode the agent interacted with a different causal DAG $\mathcal{G}$. $\mathcal{G}$ was drawn randomly from the space of possible DAGs under the constraints given in the next paragraph. Each episode consisted of $T$ steps, and was divided into two phases: *information* and *quiz*. The information phase, corresponding to the first $T-1$ steps, allowed the agent to collect information by interacting with or passively observing samples from $\mathcal{G}$. The agent could potentially use this information to infer the connectivity and weights of $\mathcal{G}$. The quiz phase, corresponding to the final step $T$, required the agent to exploit the causal knowledge it collected in the information phase, to select the node with the highest value under a random external intervention.

**Causal graphs, observations, and actions.** We generated all graphs on $N = 5$ nodes, with edges only in the upper triangular of the adjacency matrix (this guarantees that all the graphs obtained are DAGs), with edge weights, $w_{ji} \in \{-1, 0, 1\}$ (uniformly sampled), and removed 300 for held-out testing. The remaining 58749 (or $3^{N(N-1)/2} - 300$) were used as the training set. Each node's value, $X_i \in \mathbb{R}$, was Gaussian-distributed. The values of parentless nodes were drawn from $\mathcal{N}(\mu = 0.0, \sigma = 0.1)$. The conditional probability of a node with parents was $p(X_i | \mathrm{pa}(X_i)) = \mathcal{N}(\mu = \sum_j w_{ji} X_j, \sigma = 0.1)$, where $\mathrm{pa}(X_i)$ represents the parents of node $X_i$ in $\mathcal{G}$. The values of the 4 observable nodes (the root node, was always hidden), were concatenated to create $v_t$ and provided to the agent in its observation vector, $o_t = [v_t, m_t]$, where $m_t$ is a one-hot vector indicating external intervention during the quiz phase (explained below).[4]

In both phases, on each step, $t$, the agent's action, $a_t$, was a discrete choice from the range $\{1...2(N-1)\}$. Action choices in $\{1...N-1\}$ corresponded to *information actions*, and choices in $\{N...2(N-1)\}$ corresponded to *quiz actions*.

---

[4] While a simple domain provides the most unencumbered test for causal reasoning, we also carried out simulations with more complex causal graphs (graphs with non-linear connections, and larger graphs of size N = 6) and stronger requirements for generalization (holding-out entire equivalence classes of causal graphs from training) to demonstrate the robustness of our approach (see Appendix).

**Information phase.** In the information phase, an information action, $a_t$, caused an intervention on the $a_t$-th node, setting its value to $X_{a_t} = 5$. We choose an intervention value outside the likely range of sampled observations, to facilitate learning of the causal graph. The observation from the intervened graph, $\mathcal{G}_{\to X_{a_t}=5}$, was sampled similarly to $\mathcal{G}$, except the incoming edges to $X_{a_t}$ were severed, and its intervened value was used for conditioning its children's values. The node values in $\mathcal{G}_{\to X_{a_t}=5}$ were distributed as $p_{\to X_i=5}(X_{1:N\setminus i}|X_i=5)$. If a quiz action was chosen during the information phase, it was ignored, the $\mathcal{G}$ values were sampled as if no intervention had been made, and the agent was given a penalty of $r_t = -5$ in order to encourage it to take quiz actions at only during quiz phase. After the action was selected, an observation was provided to the agent. The default length of this phase was fixed to $T = N = 5$ since in the noise-free limit, a minimum of $T-1 = 4$ interventions are required in general to resolve the causal structure, and score perfectly on the test phase.

**Quiz phase.** In the quiz phase, one non-hidden node was selected at random to be intervened on externally, $X_j$, and its value was set to $-5$. We chose an intervention value of $-5$ never previously observed by the agent in that episode, thus disallowing the agent from memorizing the results of interventions in the information phase to perform well on the quiz phase. The agent was informed of this by the observed $m_{T-1}$ (a one-hot vector which indicated which node would be intervened on), from the final pre-quiz phase time-step, $T-1$. Note, $m_t$ was set to a zero-vector for steps $t < T-1$. A quiz action, $a_T$, chosen by the agent indicated the node whose value would be given to the agent as a reward. In other words, the agent would receive reward, $r_T = X_{a_T-(N-1)}$. Again, if a quiz action was chosen during the information phase, the node values were not sampled and the agent was simply given a penalty of $r_T = -5$.

**Active vs passive agents.** Our agents had to perform two distinct tasks during the information phase: a) actively choose which nodes to set values on, and b) infer the causal DAG from its observations. We refer to this setup as the "active" condition. To control for (a), we created the "passive" condition, where the agent's information phase actions are not learned. To provide a benchmark for how well the active agent can perform task (a), we fixed the passive agent's intervention policy to be an exhaustive sweep through all observable nodes. This is close to optimal for this domain – in fact it is the optimal policy for noise-free conditional node values. We also compared the active agent's performance to a baseline agent whose policy is to intervene randomly on the observable nodes in the information phase, in the Appendix.

**Two kinds of learning** The "inner loop" of learning (see Section 2.2) occurs within each episode where the agent is learning from the evidence it gathers during the information phase in order to perform well in the quiz phase. The same agent then enters a new episode, where it has to repeat the task on a different DAG. Test performance is reported on DAGs that the agent has never previously seen, after all the weights of the RNN have been fixed. Hence, the only transfer from training to test (or the "outer loop" of learning) is the ability to discover causal dependencies based on observations in the information phase, and to perform causal inference in the quiz phase.

**Agent Architecture and Training**

We used a long short-term memory (LSTM) network (Hochreiter & Schmidhuber, 1997) (with 96 hidden units) that, at each time-step $t$, receives a concatenated vector containing $[o_t, a_{t-1}, r_{t-1}]$ as input, where $o_t$ is the observation[5], $a_{t-1}$ is the previous action (as a one-hot vector) and $r_{t-1}$ the reward (as a single real-value)[6]. The outputs, calculated as linear projections of the LSTM's hidden state, are a set of policy logits (with dimensionality equal to the number of available actions), plus a scalar baseline. The policy logits are transformed by a softmax function, and then sampled to give a selected action.

Learning was by *asynchronous advantage actor-critic* (Mnih et al., 2016). In this framework, the loss function consists of three terms – the policy gradient, the baseline cost and an entropy cost. The baseline cost was weighted by 0.05 relative to the policy gradient cost. The weighting of the entropy cost was annealed over the course of training from 0.05 to 0. Optimization was done by RMSProp with $\epsilon = 10^{-5}$, momentum = 0.9 and decay = 0.95. Learning rate was annealed from $3 \times 10^{-6}$ to 0. For all experiments, after training, the agent was tested with the learning rate set to zero, on a held-out test set.

---

[5]'Observation' $o_t$ refers to the reinforcement learning term, i.e. the input from the environment to the agent. This is distinct from observations in the causal sense (referred to as observational data) i.e. samples from a casual structure where there is no information about interventions that have been carried out.

[6]These are both set to zero for the first step in an episode.

## 4 EXPERIMENTS

Our three experiments (observational, interventional, and counterfactual) differed in the properties of the $v_t$ that was observed by the agent during the information phase, and thereby limited the extent of causal reasoning possible within each data setting. Our measure of performance is the reward earned in the quiz phase for held-out DAGs. Choosing a random node node in the quiz phase results in a reward of $-5/4\!=\!-1.25$, since one node (the externally intervened node) always has value $-5$ and the others have on average $0$ value. By learning to simply avoid the externally intervened node, the agent can earn on average $0$ reward. Consistently picking the node with the highest value in the quiz phase requires the agent to perform causal reasoning. For each agent, we take the average reward earned across 1200 episodes (300 held-out test DAGs, with 4 possible external interventions). We train 12 copies of each agent and report the average reward earned by these, with error bars showing 95% confidence intervals.

### 4.1 EXPERIMENT 1: OBSERVATIONAL SETTING

In Experiment 1, the agent could neither intervene to set the value of variables in the environment, nor observe any external interventions. In other words, it only received observations from $\mathcal{G}$, not $\mathcal{G}_{\to X_j}$ (where $X_j$ is a node that has been intervened on). This limits the extent of causal inference possible. In this experiment, we tested six agents, four of which were learned: "Observational", "Long Observational", "Active Conditional", "Passive Conditional", "Observational MAP Baseline"(not learned) and the "Optimal Associative Baseline" (not learned). We also ran two other standard RL baselines—see the Appendix for details.

**Observational Agents**: In the information phase, the actions of the agent were ignored[7], and the observational agent always received the values of the observable nodes as sampled from the joint distribution associated with $\mathcal{G}$. In addition to the default $T\!=\!5$ episode length, we also trained this agent with $4\times$ longer episode length (Long Observational Agent), to measure performance increase with more observational data.

**Conditional Agents**: The information phase actions corresponded to observing a world in which the selected node $X_j$ is equal to $X_j\!=\!5$, and the remaining nodes are sampled from the conditional distribution $p(X_{1:N\setminus j}|X_j\!=\!5)$, where $X_{1:N\setminus j}$ indicates the set of all nodes except $X_j$. This differs from intervening on the variable $X_j$ by setting it to the value $X_j\!=\!5$, since here we take a conditional sample from $\mathcal{G}$ rather than from $\mathcal{G}_{\to X_j=5}$ (i.e. from $p_{\to X_j=5}(X_{1:N\setminus j}|X_j\!=\!5)$), and inference about the corresponding node's parents is possible. Therefore, this agent still has access to only observational data, as with the observational agents. However, on average it receives more diagnostic information about the relation between the random variables in $\mathcal{G}$, since it can observe samples where a node takes a value far outside the likely range of sampled observations. We run active and passive versions of this agent as described in Section 3

**Optimal Associative Baseline:** This baseline receives the true joint distribution $p(X_{1:N})$ implied by the DAG in that episode, therefore it has full knowledge of the correlation structure of the environment[8]. It can therefore do exact associative reasoning of the form $p(X_j|X_i\!=\!x)$, but cannot do any cause-effect reasoning of the form $p_{\to X_i=x}(X_j|X_i\!=\!x)$. In the quiz phase, this baseline chooses the node that has the maximum value according to the true $p(X_j|X_i\!=\!x)$ in that episode, where $X_i$ is the node externally intervened upon, and $x\!=\!-5$.

**Observational MAP Baseline**: This baseline follows the traditional method of separating causal induction and causal inference. We first carry out exact maximum a posteriori (MAP) inference over the space of DAGs in each episode (i.e. causal induction) by selecting the DAG ($\mathcal{G}^{\text{MAP}}$) of the $59049$ unique possibilities that maximizes the likelihood of the data observed, $v_{1:T}$, by the Observational Agent in that episode. This is equivalent to maximizing the posterior probability since the prior over graphs is uniform.

### RESULTS

We focus on three key questions in this experiment: (i) Can our agents learn to do associative reasoning with observational data?, (ii) Can they learn to do cause-effect reasoning from observational data?, and (iii) In addition to making causal inferences, can our agent also choose good actions in the information phase to generate the data it observes?

---

[7]These agents also did not receive the out-of-phase action penalties during the information phase since their actions are totally ignored.

[8]Notice that the agent does not know the graphical structure, i.e. it does not know which nodes are parents of which other nodes

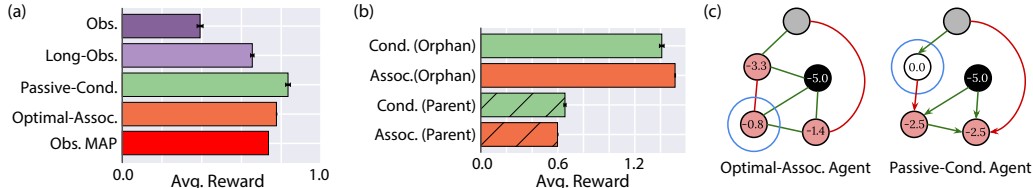

Figure 2: Experiment 1. Agents do associative and cause-effect reasoning from observational data. a) Average reward earned by the agents tested in this experiment. See main text for details. b) Performance split by the presence or absence of at least one parent (Parent and Orphan respectively) on the externally intervened node. c) Quiz phase for a test DAG. Green (red) edges indicate a weight of $+1$ $(-1)$. Black represents the intervened node, green (red) nodes indicate a positive (negative) value at that node, white indicates a zero value. The blue circles indicate the agent's choice. Left panel: $\mathcal{G}$ and the nodes taking the mean values prescribed by $p(X_{1:N\setminus j}|X_j = -5)$, including backward inference to the intervened node's parent. The Optimal Associative Baseline's choice is consistent with maximizing these (incorrect) node values. Right panel: $\mathcal{G}_{\to X_j = -5}$ and the nodes taking the mean values prescribed by $p_{\to X_j = -5}(X_{1:N\setminus j}|X_j = -5)$. We see that the Passive-Conditional Agent's choice is consistent with maximizing these (correct) node values.

For (i), we see that the Observational Agents achieve reward above the random baseline (see the Appendix), and that more observations (Long Observational Agent) lead to better performance (Fig. 2a), indicating that the agent is indeed learning the statistical dependencies between the nodes. We see that the performance of the Passive-Conditional Agent is better than either of the Observational Agents, since the data it observes is very informative about the statistical dependencies in the environment. Finally, we see that the Passive-Conditional Agent's performance is comparable (in fact surpasses as discussed below) the performance of the Optimal Associative Baseline, indicating that it is able to do perfect associative inference.

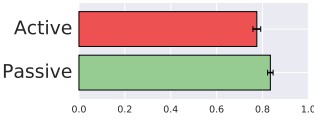

Figure 1: Active and Passive Conditional Agents

For (ii), we see the crucial result that the Passive-Conditional Agent's performance is significantly above the Optimal Associative Baseline, i.e. it performs better than what is possible using only correlations. We compare their performances, split by whether or the node that was intervened on in the quiz phase of the episode has a parent (Fig. 2b). If the intervened node $X_j$ has no parents, then $\mathcal{G} = \mathcal{G}_{\to X_j}$, and there is no advantage to being able to do cause-effect reasoning. We see indeed that the Passive-Conditional agent performs better than the Optimal Associative Baseline only when the intervened node has parents (denoted by hatched bars in Fig. 2b), indicating that this agent is able to carry out some cause-effect reasoning, despite access to only observational data – i.e. it learns some form of do-calculus. We show the quiz phase for an example test DAG in Fig. 2c, seeing that the Optimal Associative Baseline chooses according to the node values predicted by $\mathcal{G}$ whereas the Passive-Conditional Agent chooses according the node values predicted by $\mathcal{G}_{\to X_j}$.

For (iii), we see (Fig. 2) that the Active-Conditional Agent's performance is only marginally below the performance of the Passive-Conditional Agent, indicating that when the agent is allowed to choose its actions, it makes reasonable choices that allow good performance.

## 4.2 EXPERIMENT 2: INTERVENTIONAL SETTING

In Experiment 2, the agent receives interventional data in the information phase – it can choose to intervene on any observable node, $X_j$, and observe a sample from the resulting graph $\mathcal{G}_{\to X_j}$. As discussed in Section 2.1, access to intervention data permits cause-effect reasoning even in the presence of unobserved confounders, a feat which is in general impossible with access only to observational data. In this experiment, we test four new agents, two of which were learned: "Active Interventional", "Passive Interventional", "Interventional MAP Baseline"(not learned), and "Optimal Cause-Effect Baseline" (not learned).

**Interventional Agents:** The information phase actions correspond to performing an intervention on the selected node $X_j$ and sampling from $\mathcal{G}_{\to X_j}$ (see Section 3 for details). We run active and passive versions of this agent as described in Section 3.

**Interventional MAP Baseline**: This baseline infers a DAG by maximizing the likelihood of the data observed by the Passive Interventional Agent in that episode. In the quiz phase, we predict the values of

(a) (b) (c)

Figure 4: Experiment 2. Agents do cause-effect reasoning from interventional data. a) Average reward earned by the agents tested in this experiment. See main text for details. b) Performance split by the presence or absence of unobserved confounders (abbreviated as Conf. and Unconf. respectively) on the externally intervened node. c) Quiz phase for a test DAG. See Fig. 2 for a legend. Here, the left panel shows the full $\mathcal{G}$ and the nodes taking the mean values prescribed by $p(X_{1:N\setminus j}|X_j = -5)$. We see that the Passive-Cond Agent's choice is consistent with choosing based on these (incorrect) node values. The right panel shows $\mathcal{G}_{\rightarrow X_j = -5}$ and the nodes taking the mean values prescribed by $p_{\rightarrow X_j = -5}(X_{1:N\setminus j}|X_j = -5)$. We see that the Passive-Int. Agent's choice is consistent with maximizing on these (correct) node value.

each node according to $\mathcal{G}^{\text{MAP}}_{\rightarrow X_j}$ where $X_j$ is the node externally intervened upon (i.e. causal inference), and choose the node with the highest value.

**Optimal Cause-Effect Baseline:** This baseline receives the true DAG, $\mathcal{G}$. In the quiz phase, it chooses the node that has the maximum value according to $\mathcal{G}_{\rightarrow X_j}$, where $X_j$ is the node externally intervened upon.

RESULTS

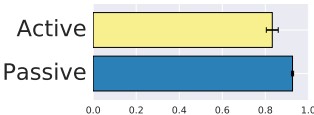

Figure 3: Active and Passive Interventional Agents

We focus on three key questions in this experiment: (i) Can our agents learn to do cause-effect reasoning from interventional data?, (ii) How does the cause-effect reasoning in our agents which have access to interventional data differ from the cause-effect reasoning measured in Experiment 1 (in agents that have access only to observational data)? (iii) In addition to making causal inferences, can our agent also choose good actions in the information phase to generate the data it observes?

For (i) we see in Fig. 4a that the Passive-Interventional Agent's performance is comparable to the Optimal Cause-Effect Baseline, indicating that it is able to do close to perfect cause-effect reasoning in this domain.

For (ii) we see in Fig. 4a the crucial result that the Passive-Interventional Agent's performance is significantly better than the Passive-Conditional Agent. We compare the performances of these two agents, split by whether the node that was intervened on in the quiz phase of the episode had unobserved confounders with other variables in the graph (Fig. 4b). In confounded cases, as described in Section 2.1, cause-effect reasoning is impossible with only observational data. We see that the performance of the Passive-Interventional Agent does not vary significantly with confoundedness, whereas the performance of the Passive-Conditional Agent is significantly lower in the confounded cases. This indicates that the improvement in the performance of the agent that has access to interventional data (as compared to the agents that had access to only observational data) is largely driven by its ability to also do cause-effect reasoning in the presence of confounders. This is highlighted by Fig. 4c, which shows the quiz phase for an example DAG, where the Passive-Conditional agent is unable to resolve the confounder, but the Passive-Interventional agent can.

For (iii), we see in Fig. 3 that the Active-Interventional Agent's performance is only marginally below the performance of the near optimal Passive-Interventional Agent, indicating that when the agent is allowed to choose its actions, it makes reasonable choices that allow good performance.

### 4.3 EXPERIMENT 3: COUNTERFACTUAL SETTING

In Experiment 3, the agent was again allowed to make interventions as in Experiment 2, but in this case the quiz phase task entailed answering a counterfactual question. We explain here what a counterfactual question in this domain looks like. Consider the conditional distribution $p(X_i|\text{pa}(X_i)) = \mathcal{N}(\sum_j w_{ji}X_j, 0.1)$ as described in Section 3 as $X_i = \sum_j w_{ji}X_j + \epsilon$ where $\epsilon$ is distributed as $\mathcal{N}(0.0, 0.1)$, and represents the specific randomness introduced when taking one sample from the DAG. After observing the nodes $X_{1:N}$

Figure 5: Experiment 3. Agents do counterfactual reasoning. a) Average reward earned by the agents tested in this experiment. See main text for details. b) Performance split by if the maximum node value in the quiz phase is degenerate (Deg.) or distinct (Dist.). c) Quiz phase for an example test-DAG. See Fig. 2 for a legend. Here, the left panel shows $\mathcal{G}_{\rightarrow X_j=-5}$ and the nodes taking the mean values prescribed by $p_{\rightarrow X_j=-5}(X_{1:N\setminus j}|X_j=-5)$. We see that the Passive-Int. Agent's choice is consistent with maximizing on these node values, where it makes a random choice between two nodes with the same value. The right panel panel shows $\mathcal{G}_{\rightarrow X_j=-5}$ and the nodes taking the exact values prescribed by the means of $p_{\rightarrow X_j=-5}(X_{1:N\setminus j}|X_j=-5)$, combined with the specific randomness inferred from the previous time step. As a result of accounting for the randomness, the two previously degenerate maximum values are now distinct. We see that the Passive-CF. agent's choice is consistent with maximizing on these node values.

in the DAG in one sample, we can infer this specific randomness $\epsilon_i$ for each node $X_i$ (i.e. *abduction* as described in the Appendix) and answer counterfactual questions like "What would the values of the nodes be, had $X_j$ *in that particular sample* taken on a different value than what we observed?", for any of the nodes $X_j$. We test 2 new learned agents: "Active Counterfactual" and "Passive Counterfactual".

**Counterfactual Agents:** This agent is exactly analogous to the Interventional agent, with the addition that the exogenous noise in the last information phase step $t=T-1$ (where say $X_p=+5$), is stored and the same noise is used in the quiz phase step $t=T$ (where say $X_f=-5$). While the question our agents have had to answer correctly so far in order to maximize their reward in the quiz phase was "Which of the nodes $X_{1:N\setminus j}$ will have the highest value when $X_f$ is set to $-5$?", in this setting, we ask "Which of the nodes $X_{1:N\setminus j}$ would have had the highest value in the last step of the information phase, if instead of having $X_p=+5$, we had $X_f=-5$?". We run active and passive versions of this agent as described in Section 3.

**Optimal Counterfactual Baseline:** This baseline receives the true DAG and does exact abduction based on the exogenous noise observed in the penultimate step of the information phase, and combines this correctly with the appropriate interventional inference on the true DAG in the quiz phase.

RESULTS

We focus on two key questions in this experiment: (i) Can our agents learn to do counterfactual reasoning?, (ii) In addition to making causal inferences, can our agent also choose good actions in the information phase to generate the data it observes?

For (i), we see that the Passive-Counterfactual Agent achieves higher reward than the Passive-Interventional Agent and the Optimal Cause-Effect Baseline. To evaluate whether this difference results from the agent's use of abduction (see the Appendix for details), we split the test set into two groups, depending on whether or not the decision for which node will have the highest value in the quiz phase is affected by exogenous noise, i.e. whether or not the node with the maximum value in the quiz phase changes if the noise is resampled. This is most prevalent in cases where the maximum expected reward is degenerate, i.e. where several nodes give the same maximum reward (denoted by hatched bars in Figure 5b). Here, agents with no access to the noise have no basis for choosing one over the other, but different noise samples can give rise to significant differences in the actual values that these degenerate nodes have. We see indeed that there is no difference in the rewards received by the Passive-Counterfactual

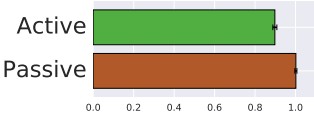

Figure 6: Active and Passive Counterfactual Agents

and Passive-Interventional Agents in the cases where the maximum values are distinct, however the Passive-Counterfactual Agent significantly outperforms the Passive-Interventional Agent in cases where there are degenerate maximum values.

For (ii), we see in Fig. 6 that the Active-Counterfactual Agent's performance is only marginally below the performance of the Passive-Counterfactual agent, indicating that when the agent is allowed to choose its actions, it makes reasonable choices that allow good performance.

## 5    Summary of results

We introduced and tested a framework for learning causal reasoning in various data settings—observational, interventional, and counterfactual—using deep meta-RL. Crucially, our approach did not require explicit encoding of formal principles of causal inference. Rather, by optimizing an agent to perform a task that depended on causal structure, the agent learned implicit strategies to use the available data for causal reasoning, including drawing inferences from passive observation, actively intervening, and making counterfactual predictions. Below, we summarize the keys results from each of the three experiments.

In Section 4.1 and Fig. 2, we show that the agent learns *to perform do-calculus*. In Fig. 2(a) we see that, compared to the highest possible reward achievable without causal knowledge, the trained agent received more reward. This observation is corroborated by Fig. 2(b) which shows that performance increased selectively in cases where do-calculus made a prediction distinguishable from the predictions based on correlations. These are situations where the externally intervened node had a parent – meaning that the intervention resulted in a different graph.

In Section 4.2 and Fig. 4, we show that the agent learns *to resolve unobserved confounders using interventions* (a feat impossible with only observational data). In Fig. 4(a) we see that the agent with access to interventional data performs better than an agent with access to only observational data. Fig. 4(b) shows that the performance increase is greater in cases where the intervened node shared an unobserved parent (a confounder) with other variables in the graph. In this section we also compare the agent's performance to a MAP estimate of the causal structure and find that the agent's performance matches it, indicating that the agent is indeed doing close to optimal causal inference.

In Section 4.3 and Fig. 5, we show that the agent learns *to use counterfactuals*. In Fig. 5(a) we see that the agent with additional access to the specific randomness in the test phase performs better than an agent with access to only interventional data. In Fig. 5(b), we find that the increased performance is observed only in cases where the maximum mean value in the graph is degenerate, and optimal choice is affected by the exogenous noise – i.e. where multiple nodes have the same value on average and the specific randomness can be used to distinguish their actual values in that specific case.

## 6    Discussion and future work

This work is the first demonstration that causal reasoning can arise out of model-free reinforcement learning. This opens up the possibility of leveraging powerful learning-based methods for causal inference in complex settings. Traditional formal approaches usually decouple the two problems of *causal induction* (i.e. inferring the structure of the underlying model) and *causal inference* (i.e. estimating causal effects and answering counterfactual questions), and despite advances in both (Ortega & Stocker, 2015; Bramley et al., 2017; Parida et al., 2018; Sen et al., 2017; Forney et al., 2017; Lattimore et al., 2016), inducing models often requires assumptions that are difficult to fit to complex real-world conditions. By learning these end-to-end, our method can potentially find representations of causal structure best tuned to the specific causal inferences required. Another key advantage of our meta-RL approach is that it allows the agent to learn to interact with the environment in order to acquire necessary observations in the service of its task—i.e. to perform active learning. In our experimental domain, our agents' active intervention policy was close to optimal, which demonstrates the promise of agents that can learn to experiment on their environment and perform rich causal reasoning on the observations.

Future work should explore agents that perform experiments to support structured exploration in RL, and optimal experiment design in complex domains where large numbers of blind interventions are prohibitive. To this end, follow-up work should focus on scaling up our approach to larger environments, with more complex causal structure and a more diverse range of tasks. Though the results here are a first step in this direction which use relatively standard deep RL components, our approach will likely benefit from more advanced architectures (e.g. Espeholt et al., 2018; Hessel et al., 2018; Hester et al., 2017) that allow longer more complex episodes, as well as models which are more explicitly compositional (e.g. Battaglia et al., 2018; Andreas et al., 2016) or have richer semantics (e.g. Ganin et al., 2018), that more explicitly leverage symmetries like equivalance classes in the environment.

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

## A    ADDITIONAL BASELINES

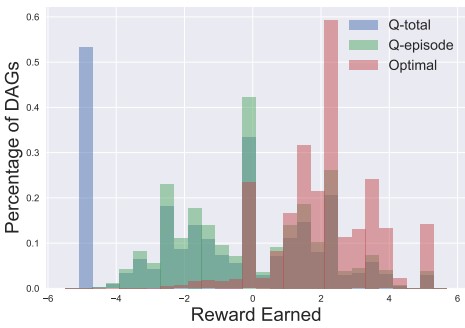

Figure 7: Reward distribution

We can also compare the performance of these agents to two standard model-free RL baselines. The Q-total agent learns a Q-value for each action across all steps for all the episodes. The Q-episode agent learns a Q-value for each action conditioned on the input at each time step $[o_t, a_{t-1}, r_{t-1}]$, but with no LSTM memory to store previous actions and observations. Since the relationship between action and reward is random between episodes, Q-total was equivalent to selecting actions randomly, resulting in a considerably negative reward. The Q-episode agent essentially makes sure to not choose the arm that is indicated by $m_t$ to be the external intervention (which is assured to be equal to $-5$), and essentially chooses randomly otherwise, giving an average reward of $0$.

## B    FORMAL DESCRIPTION OF META-LEARNING

Consider a distribution $\mathcal{D}$ over Markov Decision Processes (MDPs). We train an agent with memory (in our case an RNN-based agent) on this distribution. In each episode, we sample a task $m \sim \mathcal{D}$. At each step $t$ within an episode, the agent sees an observation $o_t$, executes an action $a_t$, and receives a reward $r_t$. Both $a_{t-1}$ and $r_{t-1}$ are given as additional inputs to the network. Thus, via the recurrence of the network, each action is a function of the entire trajectory $\mathcal{H}_t = \{o_0, a_0, r_0, ..., o_{t-1}, a_{t-1}, r_{t-1}, o_t\}$ of the episode. Because this function is parameterized by the neural network, its complexity is limited only by the size of the network.

## C    ABDUCTION-ACTION-PREDICTION METHOD FOR COUNTERFACTUAL REASONING

Pearl et al. (2016)'s "abduction-action-prediction" method prescribes one method for answering counterfactual queries, by estimating the specific unobserved makeup of individual $i$ and by transferring it to the counterfactual world. Assume, for example, the following model for $\mathcal{G}$ of Section 2.1: $E = w_{AE}A + \eta$, $H = w_{AH}A + w_{EH}E + \epsilon$, where the weights $w_{ij}$ represent the known causal effects in $\mathcal{G}$ and $\epsilon$ and $\eta$ are terms of (e.g.) Gaussian noise that represent the unobserved randomness in the makeup of each individual[9]. Suppose that for individual $i$ we observe: $A = a^i$, $E = e^i$, $H = h^i$. We can answer the counterfactual question of "What if individual $i$ had done more exercise, i.e. $E = e'$, instead?" by: a) *Abduction:* estimate the individual's specific makeup with $\epsilon^i = h^i - w_{AH}a^i - w_{EH}e^i$, b) *Action:* set $E$ to more exercise $e'$, c) *Prediction:* predict a new value for cardiac health as $h' = w_{AH}a^i + w_{EH}e' + \epsilon^i$.

## D    EXPERIMENT 4: NON-LINEAR CAUSAL GRAPHS

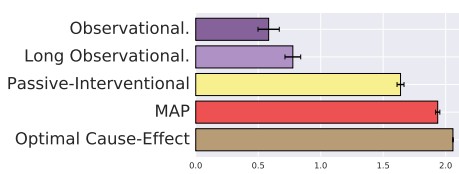

Figure 8: Experiment 4 results

The purview of the previous experiments was to show a proof of concept on a simple tractable system, demonstrating that causal induction and inference can be learned and implemented via a meta-learned agent. In this experiment, we generalize some of the results to nonlinear, non-Gaussian causal graphs which are more typical of real-world causal graphs and to demonstrate that our results hold without loss of generality on such systems.

Here we investigate causal DAGs with a quadratic dependence on the parents by changing the conditional distribution to $p(X_i | \text{pa}(X_i)) = \mathcal{N}(\frac{1}{N_i} \sum_j w_{ji}(X_j + X_j^2), \sigma)$. Here, although each node is normally

---

[9]These are zero in expectation, so without access to their value for an individual we simply use $\mathcal{G}$: $E = w_{AE}A$, $H = w_{AH}A + w_{EH}E$ to make causal predictions.

distributed given its parents, the joint distribution is not multivariate Gaussian due to the non-linearity in how the means are determined. We find that the Long-Observational achieves more reward than the Observational agent indicating that the agent is in fact learning the statistical dependencies between the nodes, within an episode. We also find that although the Active-Interventional agent is not far behind the performance of the MAP baseline, and achieves reward well above the Long-Observational[10] The fact that the MAP baseline gets so close to the Optimal Cause-Effect baseline indicates that the Active agent is choosing close to optimal actions.

# E EXPERIMENT 5: LARGER CAUSAL GRAPHS WITH GENERALIZATION TO NEW EQUIVALENCE CLASSES

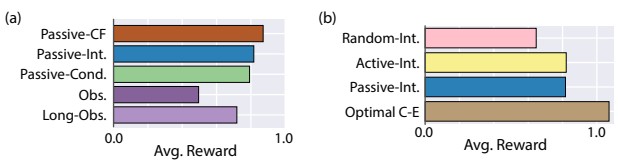

Figure 9: (a) Comparing agent performances with different data. (b) Comparing information phase intervention policies.

In the experiments reported in the main paper, the test set was a random subset of all graphs, and training examples were generated randomly subject to the constraint that they not be in the test set. However, this raised the possibility that any test graph might have an *equivalent* graph in the training set, which could result in a type of overfitting. We therefore ran a new set of experiments where the entire equivalence class of each test graph was held out from the training set[11]. Performance on the test set therefore indicates generalization of the inference procedures learned to previously unseen equivalence classes of causal DAGs. For these experiments, we used graphs with $N=6$ nodes, because 5-node graphs have too few equivalence classes to partition in this way. All other details were the same as in the main paper.

We see in Fig. 9a that the agents learn to generalize well to these held out examples, and we find the same pattern of behavior noted in the main text where the rewards earned are ordered such that Observational agent < Passive-Conditional agent < Passive-Interventional agent < Passive-Counterfactual agent. We see additionally in Fig. 9b that the Active-Interventional agent performs at par with the Passive-Interventional agent (which is allowed to see the results of interventions on all nodes) and significantly better than an additional baseline we use here of the Random-Interventional agent whose information phase policy is to intervene on nodes at random, indicating that the intervention policy learned by the Active agent is good.

# F GRAPHICAL MODELS AND BELIEF NETWORKS

Graphical models (Pearl, 1988; Bishop, 2006; Koller & Friedman, 2009; Barber, 2012; Murphy, 2012) are a marriage between graph and probability theory that allows to graphically represent and assess statistical dependence. In the following sections, we give some basic definitions and describe a method (*d-separation*) for graphically assessing statistical independence in belief networks.

BASIC DEFINITIONS

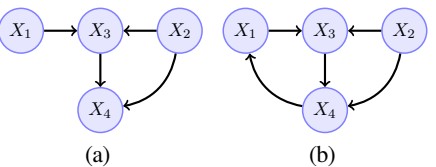

Figure 10: (a): Directed acyclic graph. The node $X_3$ is a collider on the path $X_1 \to X_3 \leftarrow X_2$ and a non-collider on the path $X_2 \to X_3 \to X_4$. (b): Cyclic graph obtained from (a) by adding a link from $X_4$ to $X_1$.

A **graph** is a collection of nodes and links connecting pairs of nodes. The links may be directed or undirected, giving rise to **directed** or **undirected graphs** respectively.

A **path** from node $X_i$ to node $X_j$ is a sequence of linked nodes starting at $X_i$ and ending at $X_j$. A **directed path** is a path whose links are directed and pointing from preceding towards following nodes in the sequence.

---

[10]The conditional distribution $p(X_{1:N \setminus j}|X_j = 5)$, and therefore Conditional agents, were non-trivial to calculate for the quadratic case.

[11]The hidden node was guaranteed to be a root node by rejecting all DAGs where the hidden node has parents

A **directed acyclic graph (DAG)** is a directed graph with no directed paths starting and ending at the same node. For example, the directed graph in Fig. 10(a) is acyclic. The addition of a link from $X_4$ to $X_1$ gives rise to a cyclic graph (Fig. 10(b)).

A node $X_i$ with a directed link to $X_j$ is called **parent** of $X_j$. In this case, $X_j$ is called **child** of $X_i$.

A node is a **collider** on a specified path if it has (at least) two parents on that path. Notice that a node can be a collider on a path and a non-collider on another path. For example, in Fig. 10(a) $X_3$ is a collider on the path $X_1 \to X_3 \leftarrow X_2$ and a non-collider on the path $X_2 \to X_3 \to X_4$.

A node $X_i$ is an **ancestor** of a node $X_j$ if there exists a directed path from $X_i$ to $X_j$. In this case, $X_j$ is a **descendant** of $X_i$.

A **graphical model** is a graph in which nodes represent random variables and links express statistical relationships between the variables.

A **belief network** is a directed acyclic graphical model in which each node $X_i$ is associated with the conditional distribution $p(X_i|\text{pa}(X_i))$, where $\text{pa}(X_i)$ indicates the parents of $X_i$. The joint distribution of all nodes in the graph, $p(X_{1:N})$, is given by the product of all conditional distributions, i.e.

$$p(X_{1:N}) = \prod_{i=1}^{N} p(X_i|\text{pa}(X_i)).$$

ASSESSING STATISTICAL INDEPENDENCE IN BELIEF NETWORKS

Given the sets of random variables $\mathcal{X}, \mathcal{Y}$ and $\mathcal{Z}$, $\mathcal{X}$ and $\mathcal{Y}$ are statistically independent given $\mathcal{Z}$ ($\mathcal{X} \perp\!\!\!\perp \mathcal{Y}|\mathcal{Z}$) if all paths from any element of $\mathcal{X}$ to any element of $\mathcal{Y}$ are **closed** (or **blocked**). A path is closed if at least one of the following conditions is satisfied:

- (Ia) There is a non-collider on the path which belongs to the conditioning set $\mathcal{Z}$.
- (Ib) There is a collider on the path such that neither the collider nor any of its descendants belong to the conditioning set $\mathcal{Z}$.

