# OpenReview forum: "Causal Reasoning from Meta-reinforcement learning"
_ICLR.cc/2019/Conference_

### Official Review · AnonReviewer1 · 2018-10-29
**Good paper on important topic**

**Rating:** 7
**Confidence:** 4

**Review:**

This submission is an great ablation study on the capabilities of modern reinforcement learning to discover the causal structure of a synthetic environment. The study separates cases where the agents can only observe or they can also act, showing the expected gains of active intervention.

The experiments are so far synthetic, but it would be really interesting to see how the lessons learned extend to more realistic environments. It would also be very nice to have a sequence of increasingly complex synthetic environments where causal inference is the task of interest, such that we can compare the performance of different RL algorithms in this task (the authors only used one).

I would change the title to "Causal Reasoning from Reinforcement Learning", since "meta-learning" is an over-loaded term and I do not clearly see its prevalence on this submission.

---

> ### Author Response · Authors · 2018-11-15
> **Reply from authors, thank you for the review.**
>
> We thank the reviewer for their interest and time. Please find below our responses to their suggestions and comments.
>
> >>The experiments are so far synthetic, but it would be really interesting to see how the lessons learned extend to more realistic environments. It would also be very nice to have a sequence of increasingly complex synthetic environments where causal inference is the task of interest, such that we can compare the performance of different RL algorithms in this task (the authors only used one).
> --------------------------------------------------
> We strongly agree that scaling our method to complex and realistic environments is worthwhile. However, we think it is outside the scope of this paper. Our goal here is to determine whether it is possible to learn a causally-aware algorithm through model-free reinforcement learning. For this purpose, a simple environment -- and correspondingly simple agent architecture -- facilitates interpretation. We therefore think it is important to start here. The current results already stretch the page limit, so we plan to follow up with scaling results in a separate paper.
>
> >>I would change the title to "Causal Reasoning from Reinforcement Learning", since "meta-learning" is an over-loaded term and I do not clearly see its prevalence on this submission.
> --------------------------------------------------
> We thank the reviewer for this great suggestion, and will accordingly update our title.
>
> We’d like to thank the reviewer again for their time and encouragement, and look forward to hearing back!

---

### Official Review · AnonReviewer2 · 2018-11-03
**Promising but several shortcomings**

**Rating:** 4
**Confidence:** 4

**Review:**


The paper presents a meta-learning RL framework to train agents that
can learn and do causal reasoning.  The paper sets up three tasks for
agents to learn to do associational, interventional, and
counterfactual reasoning. The training/testing is done on all
5-variable graphs. The authors demonstrate how the agent can maximize
their rewards, which demonstrate that the agent might have learnt to
learn some causal structure and do reasoning in the data.

Review:

I think Causality is an important area, and seeing how RL can help in
any aspect is something really worth looking into.

However, I have a few qualms about the setting and the way the tasks
are modeled.

1. Why is the task to select the node with the highest "value"
(value=expected value?  the sample? what is it?) under some random
external intervention? It feels very indirect.

Why not explicitly model certain useful actions that directly query
the structure, such as:

- selecting nodes that are parents/children of a node
- evaluating p(x | y) or p(x | do(y))?

2. The way you generate test data might introduce biases:

- If you enumerate 3^(n(n-1)/2) DAGs, some of them will have loops.  Do you weed them out?
  Does it matter?

- How do you sample weights from {-1, 0, 1}? uniform?  What happens if
  wij = 0?  This introduces bias in your training data.  This means
  your distribution is over DAGs + weights, not just DAGs.

- Your training/test split doesn't take into account certain
  equivalence/symmetries that might be present in your training data,
  making it hard to rule out whether your agents are in effect
  memorizing training data, specially that the number of test graphs
  is so tiny (300, while test could have been in the thousands too):

Example, if you graph just has one causal connection with weight = 1:
  X1 -> X2; X3; X4; X5, This is clearly equivalent to X2 -> X1; X3; X4; X5.
  Or the structure X1 -> X2 might be present in a larger graph, example with these two components:
  X1 -> X2; X3 -> X4 -> X5;

3. Why such a low number of learning steps T (T=5 in paper) in each episode? no
experimentation over choice of T or discussion of this choice is
given.  And it is mentioned in the findings, in several cases, that
the active agent is only merely comparable to the passive agent, while
one would think active would be better. If T were reasonably higher
(not too low, not too high), one would expect to see a difference.

4. Although I have concerns listed above, something about Figure 2(a)
  seems to suggest that the agent is learning something.  I think if
  you had tried to probe into what the agent is actually learning, it
  would have clarified many doubts.

However, in Figure 2(c), if the black node is -5, why is the node
below left at -2.5?  The weight on the edge is 1 and other parent is
0, so -2.5 seems extremely unlikely, given that the variance is 0.1
(stdev ~ 0.3, so ~8 standard deviations away!).  (Similar issue in
Figure 3c)

5. Although the random choice would result in a score of -5/4, I think
  it's quite easy and trivial to beat that by just ignoring the node
  that's externally intervened on and assigned -5, given it's a small
  value. This probably doesn't require the agent to be able to do
  "causal reasoning" ...  That immediately gives you a lower bound of
  0.  That might be more appropriate.

  If you could give a distribution of the max(mean_i(Xi)) over all
  graphs (including weights in your distribution), it could give an
  idea of how unlikely it is for the agent to get a high score without
  actually learning the causal structure.

Suggestions for improving the work:

- Provide results on wider range of experiments (eg more even
  train-test split, choice of T), or at minimum justify choices
  made. And address the issues above.

- Focus on more intuitive notions that clearly require causal
  knowledge, or motivate your objective very clearly to show its
  sufficiency.

- Perhaps discuss simpler examples (e.g., 3 node), where it's easy to
  enumerate all causal structures and group them into appropriate
  equivalence classes.

- Please proof-read and make sure you've defined all terms (there are
  a few, such as Xp/Xf in Expt 3, where p/f are not really defined).

- You could show a few experiments with larger N by sampling from the space of all possible
  DAGs, instead of enumerating everything.

Of course, it would be great if you can probe the agent to see what it
really learnt. But I understand that could be a long-shot.

Another open problem  is whether this approach can scale to larger number of
variables, in particular the learning might be very data hungry.

---

> ### Author Response · Authors · 2018-11-15
> **Reply from authors, thank you for the review (1/3)**
>
> TL;DR: We will significantly improve clarity of our paper and will update our results with larger graphs, addressing the reviewers’ concerns about train/test splits and scalability.
>
> We thank the reviewer for their detailed review! We think all of the points are readily addressable without fundamental changes to the paper. We hope that the reviewer will find our paper much improved based on the changes and clarifications detailed below.
> --------------------------------------------------
>
> >>1. Why is the task to select the node with the highest "value"
> (value=expected value?  the sample? what is it?) under some random
> external intervention? It feels very indirect.
>
> >>Why not explicitly model certain useful actions that directly query
> the structure, such as:
>
> >>- selecting nodes that are parents/children of a node
> >>- evaluating p(x | y) or p(x | do(y))?
>
> >>The agent's reward was the value of the chosen node in the sample at that time step.
>
> We did consider training the agent to perform explicit causal inference, but instead choose this more indirect objective to demonstrate that RL algorithms can learn to infer and utilize underlying causal structure when it is relevant to the rewarding task even when the task does not explicitly involve resolving that causal structure. This allows us to make a more general statement that also applies to the kinds of tasks prevalent in RL.
>
> --------------------------------------------------
>
> >>2. The way you generate test data might introduce biases:
>
> >>- If you enumerate 3^(n(n-1)/2) DAGs, some of them will have loops.  Do you weed them out?
>   Does it matter?
>
> >>- How do you sample weights from {-1, 0, 1}? uniform?  What happens if
>   wij = 0?  This introduces bias in your training data.  This means
>   your distribution is over DAGs + weights, not just DAGs.
>
> >>- Your training/test split doesn't take into account certain
>   equivalence/symmetries that might be present in your training data,
>   making it hard to rule out whether your agents are in effect
>   memorizing training data, specially that the number of test graphs
>   is so tiny (300, while test could have been in the thousands too):
>
> >>Example, if you graph just has one causal connection with weight = 1:
>   X1 -> X2; X3; X4; X5, This is clearly equivalent to X2 -> X1; X3; X4; X5.
>   Or the structure X1 -> X2 might be present in a larger graph, example with these two components:
>   X1 -> X2; X3 -> X4 -> X5;
>
> Our sampling procedure was not adequately explained in our paper; we thank the reviewer for drawing our attention to our lack of clarity on this point. We have tried to delineate the process more clearly below, and will include an improved explanation in the updated manuscript.
>
> We first consider the n*(n-1)/2 edges represented by the upper-diagonal of the adjacency matrix. Any graph that only contains only some subset of these edges is guaranteed to be a Directed Acyclic Graph, and contains no loops. The number 3^(n*(n-1)/2) for the total number of graphs is derived from each of these edges independently having weights -1, 0, or 1. As the reviewer pointed out, we are indeed sampling from a distribution of DAGs + weights. We do not uniformly sample over the space of DAGs, but rather we sample uniformly over the space of graphs formed by randomly assigning the edges in the upper triangular of the adjacency matrix uniformly from {0, -1, 1}. These are all guaranteed to be DAGs. Our sampling procedure means that nodes are more likely to be connected than if we had sampled the presence or absence of each edge uniformly, but this does not affect our results.
>
> The observation about equivalence classes is an excellent point. We would like to point out however, that while such equivalences exist, they are not obvious to a neural network, and the examples outlined above by the reviewer cannot be solved with memorization. Nevertheless, we agree that generalization outside the equivalence class is a stronger claim and we will update our results with simulations where we exclude entire equivalence classes from the training set and test on these held-out classes.
>
> --------------------------------------------------
>
> continued...

---

> > ### Author Response · Authors · 2018-11-15
> > **Reply from authors, thank you for the review (2/3)**
> >
> > >>3. Why such a low number of learning steps T (T=5 in paper) in each episode? no
> > experimentation over choice of T or discussion of this choice is
> > given.  And it is mentioned in the findings, in several cases, that
> > the active agent is only merely comparable to the passive agent, while
> > one would think active would be better. If T were reasonably higher
> > (not too low, not too high), one would expect to see a difference.
> >
> > We thank the reviewer for this useful feedback. We choose an episode length of T = 5 (= number of nodes) with 4 learning steps and 1 test step since, in the noise-free limit, exactly 4 interventions (one on each of the 4 observable nodes) are sufficient to fully distill the causal structure required to get maximum performance on the test phase (since the hidden node is never intervened upon in the test phase). In a longer episode, training would be more difficult since the reward (only given once at the end of each episode) becomes sparser and credit assignment more challenging [1]. Meanwhile, in a shorter episode it would be impossible to infer the causal structure in general.
> >
> > To clarify the relative performances of the passive and active agents: The intervention policy (a single intervention at each of the 4 observable nodes) of the passive agent is a good policy for this domain. In the limit of zero noise, as mentioned above, it is the optimal intervention policy. The active agent performs worse because in addition to learning to reason from the results of interventions, it must also learn an exploration policy.  Given a passive agent with a suboptimal fixed policy, in a task where smart exploration is crucial, indeed we expect the passive agent to perform worse than an agent that can actively learn an exploration policy. However, in this work, the passive policy is close to optimal and therefore acts as a benchmark for the active agent.
> >
> > We will include clarification on both of these points in our updated manuscript.
> >
> > -------------------
> >
> > >>4. Although I have concerns listed above, something about Figure 2(a)
> >   seems to suggest that the agent is learning something.  I think if
> >   you had tried to probe into what the agent is actually learning, it
> >   would have clarified many doubts.
> >
> > >>However, in Figure 2(c), if the black node is -5, why is the node
> > below left at -2.5?  The weight on the edge is 1 and other parent is
> > 0, so -2.5 seems extremely unlikely, given that the variance is 0.1
> > (stdev ~ 0.3, so ~8 standard deviations away!).  (Similar issue in
> > Figure 3c)
> >
> > We appreciate the attention to detail! We did find an error in Figure 4(c) and have addressed it in the new version of the manuscript. Figure 2(c) however seems correct. The mean value of a child node is given by the weighted mean of its parents’ values.  So -2.5 = (-5.0 x 1 + 0.0 x 1)/2.
> >
> > Regarding “what the agent is actually learning”:
> >
> > In Section 4.1 and Figure 2, we show that the agent learns to perform some do-calculus. In Figure 2(a) we see that, compared to the highest possible reward achievable without causal knowledge, the trained agent received more reward. This observation is corroborated by Figure 2(b) which shows that performance increased selectively in cases where do-calculus made a prediction distinguishable from the predictions based on correlations. These are situations where the externally intervened node had a parent -- meaning that the intervention resulted in a different graph.
> >
> > In Section 4.2 and Figure 4, we show that the agent learns to resolve unobserved confounders using interventions (a feat impossible with only observational data). In Figure 4(a) we see that the agent with access to interventional data performs better than an agent with access to only observational data. Figure 4(b) shows that the performance increase is greater in cases where the intervened node shared an unobserved parent (a confounder) with other variables in the graph. In this section we also compare the agent’s performance to a MAP estimate of the causal structure and find that the agent’s performance matches it, indicating that the agent is indeed doing close to optimal causal inference.
> >
> > In Section 4.3 and Figure 6, we show that the agent learns to use counterfactuals. In Figure 6(a) we see that the agent with additional access to the specific randomness in the test phase performs better than an agent with access to only interventional data. In Figure 6(b), we find that the increased performance is observed only in cases where the maximum mean value in the graph is degenerate, and optimal choice is affected by the exogenous noise -- i.e. where multiple nodes have the same value on average and the specific randomness can be used to distinguish their actual values in that specific case.
> >
> > In all of these three cases we also show an example graph (in Figures 2(c), 4(c), and 6(c)) where the agent’s behavior demonstrates what it has learned in each experiment.
> >
> > We will make these three points more directly in the revised text.
> >
> > contd.

---

> > > ### Author Response · Authors · 2018-11-15
> > > **Reply from authors, thank you for the review (3/3)**
> > >
> > >
> > > >>5. Although the random choice would result in a score of -5/4, I think
> > >   it's quite easy and trivial to beat that by just ignoring the node
> > >   that's externally intervened on and assigned -5, given it's a small
> > >   value. This probably doesn't require the agent to be able to do
> > >   "causal reasoning" ...  That immediately gives you a lower bound of
> > >   0.  That might be more appropriate.
> > >
> > > We agree that this is a better baseline. In the original submission we described it in Appendix A. In the revision we will move it to the main text.
> > >
> > > --------------------------------------------------
> > >
> > > >>If you could give a distribution of the max(mean_i(Xi)) over all
> > >   graphs (including weights in your distribution), it could give an
> > >   idea of how unlikely it is for the agent to get a high score without
> > >   actually learning the causal structure.
> > >
> > > Thanks for this suggestion. We will add a plot showing the distributions of the values of mean_i(Xi) and max(mean_i(Xi)) over all graphs.
> > > --------------------------------------------------
> > >
> > > >>Suggestions for improving the work:
> > >
> > > We have grouped some of the suggestions below so as to make responses more concise.
> > > --------------------------------------------------
> > >
> > > >>- Focus on more intuitive notions that clearly require causal
> > >   knowledge, or motivate your objective very clearly to show its sufficiency.
> > >
> > > >>Of course, it would be great if you can probe the agent to see what it
> > > really learnt. But I understand that could be a long-shot.
> > >
> > > We have tried to clarify above our analysis of what the agent learned.
> > >
> > > --------------------------------------------------
> > >
> > > >>- Perhaps discuss simpler examples (e.g., 3 node), where it's easy to
> > >   enumerate all causal structures and group them into appropriate
> > >   equivalence classes.
> > >
> > > In the revision, we will explicitly partition equivalence classes between training and testing, as described above. We will then also show examples of 3-node graphs at test time whose equivalence class was excluded from training.
> > >
> > > --------------------------------------------------
> > >
> > > >>- Provide results on wider range of experiments (eg more even
> > >   train-test split, choice of T), or at minimum justify choices
> > >   made. And address the issues above.
> > >
> > >
> > > >>- You could show a few experiments with larger N by sampling from the space of all possible
> > >   DAGs, instead of enumerating everything.
> > >
> > > >>Another open problem  is whether this approach can scale to larger number of
> > > variables, in particular the learning might be very data hungry.
> > >
> > >
> > > We hope to have addressed the reviewers' concerns about the setup in the replies above. The goal of our paper was to show the first evidence that causal reasoning can arise from model-free RL. We also demonstrate that the same agent discovers a good experimentation policy from which it learns the environment’s causal structure. This lays the groundwork for sophisticated reinforcement learning agents that can actively interact with and learn about their environments. Our synthetic environment of 5 node random DAGs is simple enough to concretely demonstrate causal reasoning, as well as to benchmark the learned intervention policy; but is not tailored so specifically to causal inference as to lose connection to typical RL tasks. While scalability is an important question, we think that it is outside the scope of this paper.
> > >
> > >  However, in the revision, we will discuss issues related to scaling: a) compositional agent architectures might allow the agent to utilize symmetries such as equivalence classes, thus lowering the training data requirements [2], and b) more advanced training regimes might alleviate the credit assignment problem, allowing successful training with longer episode lengths [3].
> > >
> > > --------------------------------------------------
> > >
> > > >>- Please proof-read and make sure you've defined all terms (there are
> > >   a few, such as Xp/Xf in Expt 3, where p/f are not really defined).
> > >
> > > These points are duly noted and we will address them in our updated manuscript.
> > >
> > > We’d like to thank the reviewer again for their detailed and insightful comments, and look forward to their reply. Our manuscript has hugely benefited from this feedback.
> > >
> > >
> > > [1] Hochreiter, S., Bengio, Y., Frasconi, P., & Schmidhuber, J. (2001). Gradient flow in recurrent nets: the difficulty of learning long-term dependencies.
> > > [2] Battaglia, P. W., Hamrick, J. B., Bapst, V., Sanchez-Gonzalez, A., Zambaldi, V., Malinowski, M., ... & Gulcehre, C. (2018). Relational inductive biases, deep learning, and graph networks.
> > > [3] Bengio, Y., & Frasconi, P. (1994). Credit assignment through time: Alternatives to backpropagation.

---

### Official Review · AnonReviewer3 · 2018-11-14
**Potentially interesting, but possibly not ready yet**

**Rating:** 4
**Confidence:** 3

**Review:**

This paper aims at training agents to perform causal reasoning with RL in three settings: observational (the agent can only obtain one observational sample at a time), interventional (the agent can obtain an interventional sample at a time for a given perfect intervention on a given variable) and a counterfactual setting (the agent can obtains interventional samples, but the prediction is about the case in which the same noise variables were sampled, but a different intervention was performed) . In each of these settings, after T-1 steps of information gathering, the algorithm is supposed to select the node with the highest value in the last step. Different types of agents are evaluated on a limited simulated dataset, with weak and not completely interpretable results.

Pros:
-Using RL to learn causal reasoning is a very interesting and worthwhile task.
-The paper tries to systematize the comparison of different settings with different available data.


Cons:
-Task does not necessarily require causal knowledge (predict the node with the highest value in this restricted linear setting)
-Very limited experimental setting (causal graphs with 5 nodes, one of which hidden, linear Gaussian with +/- 1 coefficients, with interventions in training set always +5, and in test set always -5) and lukewarm results, that don’t seem enough for the strong claims. This is one of the easiest ways to improve the paper.
-In the rare cases in which there are some causal baselines (e.g. MAP baseline), they seem to outperform the proposed algorithms (e.g. Experiment 2)
-Somehow the “active” setting in which the agent can decide the intervention targets seems to always perform worse than the “passive” setting, in which the targets are already chosen. This is very puzzling for me, I thought that choosing the targets should improve the results...
-Seem to be missing a lot of literature on causality and bandits, or reinforcement learning (for example: https://arxiv.org/abs/1606.03203, https://arxiv.org/abs/1701.02789, http://proceedings.mlr.press/v70/forney17a.html)
-Many details were unclear to me and in general the clarity of the description could be improved

In general, I think the paper could be opening up an interesting research direction, but unfortunately I’m not sure it is ready yet.


Details:
-Abstract: “Though powerful formalisms for causal reasoning have been developed, applying them in real-world domains is often difficult because the frameworks make idealized assumptions”. Although possibly true, this sounds a bit strong, given the paper’s results. What assumptions do your agents make? At the moment the agents you presented work on an incredibly small subset of causal graphs (not even all linear gaussian models with a hidden variable…), and it’s even not compared properly against the standard causal reasoning/causal discovery algorithms...
-Footnote 1: “this formalism for causal reasoning assumes that the structure of the causal graph is known” - (Spirtes et al. 2001) present several causal discovery (here “causal induction”) methods that recover the graph from data.
-Section 2.1 “X_i is a potential cause of X_j” - it’s a cause, not potential, maybe potentially not direct.
-Section 3: 3^(N-1)/2 is not the number of possible DAGs, that’s described by this sequence: https://oeis.org/A003024. Rather that is the number of (possibly cyclic) graphs with either -1, 1 or 0 on the edges.
-“The values of all but one node (the root node, which is always hidden)” - so is it 4 or 5 nodes? Or it is that all possible DAGs on N=6 nodes one of which is hidden? I’m asking because in the following it seems you can intervene on any of the 5 nodes…
-The intervention action is to set a given node to +5 (not really clear why), while in the quiz phase (in which the agent tries to predict the node with the highest variable) there is an intervention on a known node that is set to -5 (again not clear why, but different from the interventions seen in the T-1 steps).
-Active-Conditional is only marginally below Passive-Conditional, “indicating that when the agent is allowed to choose its actions, it makes reasonable choices” - not really, it should perform better, not “marginally below”... Same for all the other settings
-Why not use the MAP baseline for the observational case?
-What data does the Passive Conditional algorithms in Experiment 2? Only observations  (so a subset of the data)?
-What are the unobserved confounders you mention in the results of Experiment 2? I thought there is only one unobserved confounder (the root node)? Where do the others come from?
-The counterfactual setting possibly lacks an optimal algorithm?

---

> ### Author Response · Authors · 2018-11-16
> **Reply from authors, thank you for the review (1/3)**
>
> TL;DR: We will significantly improve clarity of our paper and will update our results with larger graphs, addressing the reviewers’ concerns about generalizability.
>
> We thank the reviewer for their interest, time, and detailed review! We feel that most of the criticisms are not fundamental concerns about the content but rather pertain to lack of clarity in the text. We are grateful for the opportunity to improve our exposition. We have made extensive clarifications below, and hope the reviewer will find our paper much improved as a result.
>
> We have grouped some of the suggestions below so as to make responses more coherent.
>
> --------------------------------------------------
>
>
> >>-Task does not necessarily require causal knowledge (predict the node with the highest value in this restricted linear setting)
>
>
> >>-Very limited experimental setting (causal graphs with 5 nodes, one of which hidden, linear Gaussian with +/- 1 coefficients, with interventions in training set always +5, and in test set always -5) and lukewarm results, that don’t seem enough for the strong claims. This is one of the easiest ways to improve the paper.
>
> We agree that generalization to more settings is important  but we think that our setting is sufficient to answer the question posed in this work. We respectfully disagree that the simplicity of the domain means that causal knowledge is not required. That the agent successfully makes predictions about the effect of an intervention, makes correct causal inference despite the presence of a hidden confounder, and makes counterfactual predictions, are all hallmarks of causal knowledge. The agent outperforms the best possible non-causal algorithm. We also point to standard textbooks [Causal Inference in Statistics - A Primer, Judea Pearl, Madelyn Glymour, Nicholas P. Jewell, 2016] describing and analysing causal inference in exactly such linear Gaussian settings. A non-linear setting would possibly make this more challenging for the agent, but our main goal was to demonstrate that our meta-learning approach using model-free RL, can learn to exploit causal structure changing at each episode -- an entirely new area of research. We felt that this simple setting afforded the most unencumbered test for causal reasoning.
>
> Nevertheless, we agree that some demonstration of the generalizability of our approach will strengthen our paper. In Appendix D, we present results on non-linear causal graphs, and in the revision we will include results with larger graphs. We used a test intervention (-5) far outside the learning distribution (always +5) because this is a strong test for having encoded the underlying causal graph. We agree that there are many ways to generalize our approach, and have made an effort to demonstrate it with some non-linear graphs and larger numbers of nodes;  but convincingly and fairly testing the limits of this generalizability would require using more sophisticated agents and training regimes, and is orthogonal to the purview of our current work.
>
> --------------------------------------------------
>
> >>-In the rare cases in which there are some causal baselines (e.g. MAP baseline), they seem to outperform the proposed algorithms (e.g. Experiment 2)
>
> Our analyses are focused on looking for evidence that the RL agent takes advantage of causal information. The MAP baseline is an upper bound on performance. The key result for Experiment 2 is that the agent learns an important aspect of causal reasoning i.e. to resolve unobserved confounders with interventions. In Figure 4(a) we see that the agent with access to interventional data performs better than an agent with access to only observational data, reaching close to optimal MAP performance. Figure 4(b) shows that the performance increase is greater in cases where the intervened node shared an unobserved parent (a confounder) with other variables in the graph.
>
> --------------------------------------------------
>
>
> >>-Somehow the “active” setting in which the agent can decide the intervention targets seems to always perform worse than the “passive” setting, in which the targets are already chosen. This is very puzzling for me, I thought that choosing the targets should improve the results…
>
> We apologize for not explaining this adequately in the manuscript. The intervention policy hard-coded into the passive agent (a single  intervention on each of the 4 observable nodes) is  near optimal. In the zero noise limit, it is optimal. The active agent, on the other hand, must learn a good exploration policy from scratch. This is an extra challenge that results in slightly worse performance. In future work, it might be interesting to examine domains where it is difficult to hand-craft an effective exploration policy, and determine whether the meta-RL approach can find such a policy.  We will improve our explanation of this issue in the revised text.
>
> --------------------------------------------------
>
> cont’d..

---

> > ### Author Response · Authors · 2018-11-16
> > **Reply from authors, thank you for the review (2/3)**
> >
> >
> > >>-Seem to be missing a lot of literature on causality and bandits, or reinforcement learning (for example: https://arxiv.org/abs/1606.03203, https://arxiv.org/abs/1701.02789, http://proceedings.mlr.press/v70/forney17a.html)
> >
> > We thank the reviewer for these helpful references and will incorporate a better literature review in our updated manuscript.
> >
> > --------------------------------------------------
> > >>-Many details were unclear to me and in general the clarity of the description could be improved
> >
> > We hope that by addressing the reviewers concerns as detailed below, the clarity of the updated manuscript will be much improved.
> >
> > --------------------------------------------------
> >
> > >>In general, I think the paper could be opening up an interesting research direction, but unfortunately I’m not sure it is ready yet.
> >
> > >>Details:
> > >>-Abstract: “Though powerful formalisms for causal reasoning have been developed, applying them in real-world domains is often difficult because the frameworks make idealized assumptions”. Although possibly true, this sounds a bit strong, given the paper’s results. What assumptions do your agents make? At the moment the agents you presented work on an incredibly small subset of causal graphs (not even all linear gaussian models with a hidden variable…), and it’s even not compared properly against the standard causal reasoning/causal discovery algorithms…
> >
> > We agree that our paper in its current form does not speak to improving causal reasoning in real-world domains. The goal of our paper is to demonstrate the proof of principle that causal reasoning can arise out of model-free reinforcement learning. The above statement was made to motivate the potential practical value of RL algorithms that can make fast, amortized causal inferences at run time, learned end-to-end from large and high dimensional data that might be intractable for traditional causal inference algorithms. However, we agree that this statement is strong and will edit it to make sure our contributions are more accurately represented.
> >
> > --------------------------------------------------
> > >>-Footnote 1: “this formalism for causal reasoning assumes that the structure of the causal graph is known” - (Spirtes et al. 2001) present several causal discovery (here “causal induction”) methods that recover the graph from data.
> >
> > It was not our intention to dismiss the literature, but to make clear to readers that the tasks our agent performs in the three experiments do not fully equate to the three levels of causal reasoning we formalize in the section on causality, as these assume that the graph structure is known. We will change this phrasing.
> >
> > --------------------------------------------------
> >
> >
> > >>-Section 2.1 “X_i is a potential cause of X_j” - it’s a cause, not potential, maybe potentially not direct.
> >
> > We are using the definition in the book Causal Inference in Statistics - A Primer, Judea Pearl, Madelyn Glymour, Nicholas P. Jewell, 2016, page 27:  “If X is a descendant of Y, then Y is a potential cause of X (there are rare intransitive cases in which Y will not be a cause of X, which we will discuss in Part Two).”, in the interest of being as general as possible in our introductory section on causality.
> >
> > --------------------------------------------------
> >
> >
> > >>-Section 3: 3^(N-1)/2 is not the number of possible DAGs, that’s described by this sequence: https://oeis.org/A003024. Rather that is the number of (possibly cyclic) graphs with either -1, 1 or 0 on the edges.
> >
> > Our sampling procedure was not adequately explained in our paper, we thank the reviewer for drawing our attention to our lack of clarity on this point. We have tried to delineate the process more clearly below, and will include an explanation in the updated manuscript.
> >
> > We first consider the n*(n-1)/2 edges in the strictly upper-triangular part of the adjacency matrix. The number 3^(n*(n-1)/2) for the total number of graphs is derived from each of these edges independently having weights -1, 0, or 1. These are all guaranteed to be acyclic. We will revise to be more clear about what distribution of graphs we are sampling from.
> >
> > --------------------------------------------------
> >
> > cont’d...

---

> > > ### Author Response · Authors · 2018-11-16
> > > **Reply from authors, thank you for the review (3/3)**
> > >
> > >
> > > >>-“The values of all but one node (the root node, which is always hidden)” - so is it 4 or 5 nodes? Or it is that all possible DAGs on N=6 nodes one of which is hidden? I’m asking because in the following it seems you can intervene on any of the 5 nodes…
> > >
> > > The DAGs consist of a total of 5 nodes, one of which is hidden. The agent can only intervene on 4 of the nodes. We will clarify this in the updated manuscript.
> > >
> > > --------------------------------------------------
> > >
> > > >>-The intervention action is to set a given node to +5 (not really clear why), while in the quiz phase (in which the agent tries to predict the node with the highest variable) there is an intervention on a known node that is set to -5 (again not clear why, but different from the interventions seen in the T-1 steps).
> > >
> > > Thanks for pointing out that we never explained our choices for these values. We will improve our exposition in the updated manuscript.
> > >
> > > The intervention action sets a node to a value (+5) outside the likely range of passive observations. This facilitates learning the causal graph. One of the important control conditions is an agent that receives samples from the graph conditioned on one of the nodes being +5, in order to directly assess the benefits of intervention.
> > >
> > > The intervention in the quiz phase sets a node to a value never before seen. This disallows the agent from memorizing the results of its interventions in the information phase.
> > >
> > > --------------------------------------------------
> > >
> > > >>-Active-Conditional is only marginally below Passive-Conditional, “indicating that when the agent is allowed to choose its actions, it makes reasonable choices” - not really, it should perform better, not “marginally below”... Same for all the other settings
> > >
> > > We hope to have clarified this in the explanation above.
> > >
> > > --------------------------------------------------
> > >
> > > >>-Why not use the MAP baseline for the observational case?
> > >
> > > This is a good point. We will add a MAP baseline that records the optimal causal induction / discovery possible with purely observational data. However, the key observation from this experiment was that the conditional agent outperformed the optimal associative baseline. This indicates that that the conditional agent drew causal inferences from observational data (i.e., learned to perform do-calculus). This is highlighted by the fact that this improvement manifested only in test cases where do-calculus makes a prediction distinguishable from the predictions based on correlations. These are cases where the externally intervened node has a parent, so that graph surgery results in a different graph.
> > >
> > > --------------------------------------------------
> > >
> > > >>-What data does the Passive Conditional algorithms in Experiment 2? Only observations (so a subset of the data)?
> > >
> > > The Passive Conditional agent receives conditional samples from the distribution defined by the DAG. These samples are conditioned on one of the nodes having a value of +5. This is described in the subsection about conditional agents on Page 5 of the original manuscript. However, we appreciate that this description was somewhat buried. In the revision we draw more attention to this mechanism.
> > >
> > > --------------------------------------------------
> > >
> > >
> > > >>-What are the unobserved confounders you mention in the results of Experiment 2? I thought there is only one unobserved confounder (the root node)? Where do the others come from?
> > >
> > > The reviewer is right in noting that there is only one unobserved confounder. The plural was not intended to refer to multiple confounders within a single graph, however we realize that this was confusing, and we will remove the usage of the plural in the revision.
> > >
> > > --------------------------------------------------
> > >
> > > >>-The counterfactual setting possibly lacks an optimal algorithm?
> > >
> > > There is an optimal algorithm in the counterfactual setting. Thanks for drawing attention to this. In the revision we will include a baseline that records the specific exogenous noise and draws the correct counterfactual prediction from it. Notwithstanding this, the key observation in this experiment was that the counterfactual agent earned more reward than the MAP baseline. This is sufficient to infer that the agent used information about the specific exogenous noise (i.e. counterfactual inference), and not just information about the causal structure. This observation is also consistent with the fact that the performance improvement manifested only in the presence of degenerate maximum valued nodes.
> > >
> > > We’d like to thank the reviewer again for their detailed and insightful comments, and look forward to their reply. Our manuscript has hugely benefited from this feedback.

---

### Official Review · AnonReviewer4 · 2018-11-18
**Promising paper on an appealing topic, but needs a bit more work**

**Rating:** 5
**Confidence:** 4

**Review:**

Note: This review is coming in a bit late, already after one round of responses. So I write this with the benefit of having read the helpful previous exchange.

I am generally positive about the paper and the broader project. The idea of showing that causal reasoning naturally emerges from certain decision-making tasks and that modern (meta-learning) RL agents can become attuned to causal structure of the world without being explicitly trained to answer causal questions is an attractive one. I also find much about the specific paper elegant and creative. Considering three grades of causal sophistication (from conditional probability to cause-effect reasoning to counterfactual prediction) seems like the right thing to do in this setting.

Despite these positive qualities, I was confused by many of the same issues as other reviewers, and I think the paper does need some more serious revisions. Some of these are simple matters of clarification as the authors acknowledge; others, however, require further substantive work. It sounds like the authors are committed to doing some of this work, and I would like to add one more vote of encouragement. While the paper may be slightly too preliminary for acceptance at this time, I am optimistic that a future version of this paper will be a wonderful contribution.

(*) The authors say at several points that the approach “did not require explicit knowledge of formal principles of causal inference.” But there seem to be a whole of lot of causal assumptions that are critically implicit in the setup. It would be good to understand this better. In particular, the different agents are hardwired to have access to different kinds of information. The interventional agent is provided with data that the conditional agent simply doesn’t get to see. Likewise, the counterfactual agent is provided with information about noise. Any sufficiently powerful learning system will realize that (and even how) the given information is relevant to the decision-making task at hand. A lot of the work (all of the work?) seems to be done by supplying the information that we know would be relevant.

(*) Previous reviewers have already made this point - I think it’s crucial - and it’s also related to the previous concern: It is not clear how difficult the tasks facing these agents actually are, nor is it clear that solving them genuinely requires causal understanding. What seems to be shown is that, by supplying information that’s critical for the task at hand, a sufficiently powerful learning agent is able to harness that information successfully. But how difficult is this task, and why does it require causal understanding? I do think that some of the work the authors did is quite helpful, e.g., dividing the test set between the easy and hard cases (orphan / parented, unconfounded / confounded). But I do not feel I have an adequate understanding of the task as seen, so to say, from the perspective of the agent. Specifically:

(*) I completely second the worry one of the reviewers raised about equivalence classes and symmetries. The test set should be chosen more deliberately - not randomly - to rule out deflationary explanations of the agents’ purported success. I’m happy to hear that the authors will be looking more into this and I would be interested to know how the results look.

(*) The “baselines” in this paper are often not baselines at all, but rather various optimal approaches to alternative formulations of the task. I feel we need more actual baselines in order to see how well the agents of interest are doing. I don’t know how to interpret phrases like “close to perfect” without a better understanding of how things look below perfection.

As a concrete case of this, just like the other reviewers, I was initially quite confused about the passive agents and why they did better than the active agents. These are passive agents who actually get to make multiple observations, rather than baseline passive agents who choose interventions in a suboptimal way. I think it would be helpful to compare against an agent who makes the same number of observations but chooses them in a suboptimal (e.g., random) way.

(*) In relation to the existing literature on causal induction, it’s telling that implementing a perfect MAP agent in this setting is even possible. This makes me worry further about how easy these tasks are (again, provided one has all of the relevant information about the task). But it also shows that comparison with existing causal inference methods is simply inappropriate here, since those methods are designed for realistic settings where MAP inference is far from possible. I think that’s fine, but I also think it should be clarified in the paper. The point is not (at least yet) that these methods are competitors to causal inference methods that do “require explicit knowledge of formal principles of causal inference,” but rather that we have a proof-of-concept that some elementary causal understanding may emerge from typical RL tasks when agents are faced with the right kinds of tasks and given access to the right kinds of data. That’s an interesting claim on its own. The penultimate paragraph in the paper (among other passages) seems to me quite misleading on this point.

(*) One very minor question I have is why actions were softmax selected even in the quiz phase. What were the softmax parameters? And would some of the agents not perform a bit better if they maximized?

---

> ### Author Response · Authors · 2018-11-22
> **Reply from authors, thank you for the review (1/2)**
>
>
> We thank the reviewer for their comments. We appreciate their interest in our work and believe that we can address their remaining concerns with the new simulations detailed in previous responses, and the clarifications and minor changes detailed below.
>
> >>Note: This review is coming in a bit late, already after one round of responses. So I write this with the benefit of having read the helpful previous exchange.
>
> >>I am generally positive about the paper and the broader project. The idea of showing that causal reasoning naturally emerges from certain decision-making tasks and that modern (meta-learning) RL agents can become attuned to causal structure of the world without being explicitly trained to answer causal questions is an attractive one. I also find much about the specific paper elegant and creative. Considering three grades of causal sophistication (from conditional probability to cause-effect reasoning to counterfactual prediction) seems like the right thing to do in this setting.
>
> >>Despite these positive qualities, I was confused by many of the same issues as other reviewers, and I think the paper does need some more serious revisions. Some of these are simple matters of clarification as the authors acknowledge; others, however, require further substantive work. It sounds like the authors are committed to doing some of this work, and I would like to add one more vote of encouragement. While the paper may be slightly too preliminary for acceptance at this time, I am optimistic that a future version of this paper will be a wonderful contribution.
>
> >>(*) The authors say at several points that the approach “did not require explicit knowledge of formal principles of causal inference.” But there seem to be a whole of lot of causal assumptions that are critically implicit in the setup. It would be good to understand this better. In particular, the different agents are hardwired to have access to different kinds of information. The interventional agent is provided with data that the conditional agent simply doesn’t get to see. Likewise, the counterfactual agent is provided with information about noise. Any sufficiently powerful learning system will realize that (and even how) the given information is relevant to the decision-making task at hand. A lot of the work (all of the work?) seems to be done by supplying the information that we know would be relevant.
>
> This is an interesting point. To reason effectively about causality in the real world, a major challenge is isolating the information relevant to causal reasoning. But for our agents, we sidestep this challenge by providing the information needed in a digestible format.
>
> We agree that in the long run, it will be important to develop agents that extract the right information from complex streams of raw data. In the current work, though, we just wanted to test the simplest version of a hypothesis -- whether it is possible at all for model-free RL to give rise to a causally-aware learning algorithm, when given access to the information it needs.
>
> We think that by limiting the kinds of data available to the agent in different ways, we were able to demonstrate the 3 tiers of causal reasoning in the most controlled way. If all of the agents were supplied with all of the information, it would be harder to directly assess the different aspects of causal and counterfactual reasoning. We generally agree that sufficiently powerful learning systems will realize that information is relevant to a task -- however, before now it was unknown whether standard model-free RL algorithms can induce a causal reasoning algorithm, even if provided with the relevant data.
>
> --------------------------------------------------------------------
> Continued…

---

> > ### Author Response · Authors · 2018-11-22
> > **Reply from authors, thank you for the review (2/2)**
> >
> >
> > >>(*) Previous reviewers have already made this point - I think it’s crucial - and it’s also related to the previous concern: It is not clear how difficult the tasks facing these agents actually are, nor is it clear that solving them genuinely requires causal understanding. What seems to be shown is that, by supplying information that’s critical for the task at hand, a sufficiently powerful learning agent is able to harness that information successfully. But how difficult is this task, and why does it require causal understanding? I do think that some of the work the authors did is quite helpful, e.g., dividing the test set between the easy and hard cases (orphan / parented, unconfounded / confounded). But I do not feel I have an adequate understanding of the task as seen, so to say, from the perspective of the agent. Specifically:
> >
> > >>(*) I completely second the worry one of the reviewers raised about equivalence classes and symmetries. The test set should be chosen more deliberately - not randomly - to rule out deflationary explanations of the agents’ purported success. I’m happy to hear that the authors will be looking more into this and I would be interested to know how the results look.
> >
> >
> > We are currently running simulations on larger graphs and testing on held out equivalence classes, and look forward to the reviewers comments on these new results.
> >
> > --------------------------------------------------------------------
> >
> > >>(*) The “baselines” in this paper are often not baselines at all, but rather various optimal approaches to alternative formulations of the task. I feel we need more actual baselines in order to see how well the agents of interest are doing. I don’t know how to interpret phrases like “close to perfect” without a better understanding of how things look below perfection.
> >
> > >>As a concrete case of this, just like the other reviewers, I was initially quite confused about the passive agents and why they did better than the active agents. These are passive agents who actually get to make multiple observations, rather than baseline passive agents who choose interventions in a suboptimal way. I think it would be helpful to compare against an agent who makes the same number of observations but chooses them in a suboptimal (e.g., random) way.
> >
> >
> > We agree that this would be a great addition, and thank the reviewer for suggesting it. We will improve our explanation of the active vs passive agents and will also include a new baseline in our updated draft where an agent performs random interventions in the information phase.
> >
> > --------------------------------------------------------------------
> >
> > >>(*) In relation to the existing literature on causal induction, it’s telling that implementing a perfect MAP agent in this setting is even possible. This makes me worry further about how easy these tasks are (again, provided one has all of the relevant information about the task). But it also shows that comparison with existing causal inference methods is simply inappropriate here, since those methods are designed for realistic settings where MAP inference is far from possible. I think that’s fine, but I also think it should be clarified in the paper. The point is not (at least yet) that these methods are competitors to causal inference methods that do “require explicit knowledge of formal principles of causal inference,” but rather that we have a proof-of-concept that some elementary causal understanding may emerge from typical RL tasks when agents are faced with the right kinds of tasks and given access to the right kinds of data. That’s an interesting claim on its own. The penultimate paragraph in the paper (among other passages) seems to me quite misleading on this point.
> >
> > In the current work we haven't explored scalability. The indicated paragraph of our discussion section was intended to motivate the potential future practical value of RL algorithms that learn to make causal inferences end-to-end on large, high dimensional data. We will edit the paragraph to be more clear about how the present work is a proof-of-concept.
> >
> > --------------------------------------------------------------------
> >
> > >>(*) One very minor question I have is why actions were softmax selected even in the quiz phase. What were the softmax parameters? And would some of the agents not perform a bit better if they maximized?
> >
> > We refrained from maximizing when checking test performance in order to retain maximum similarity with the training phase (during which a softmax is necessary to propagate gradients through the network). The temperature parameter was not manipulated or optimized and was kept fixed to the default of 1. However, empirically, by the end of training, the network's policy outputs (i.e., the inputs to the softmax) typically had a sufficiently large scale to make choices almost deterministic.
> >
> > --------------------------------------------------------------------

---

### Author Response · Authors · 2018-12-03
**Summary of changes in the revised manuscript**


We have endeavored to address all the reviewers' concerns, and hope they will find our manuscript much improved after the additions and clarifications detailed below:

>> Changed the title to “Causality from Meta Reinforcement Learning” incorporating feedback from reviewer 1.
>> Changed phrasing in the abstract and discussion as suggested by reviewers 3 and 4.
>> Improved the explanation of our sampling procedure in response to reviewers 2 and 3.
>> Improved the explanation of the Passive agents’ performance (and why it is higher than the Active agents’) in response to reviewers 2 and 3.
>> Included 2 new baselines (MAP for observational data, and Optimal counterfactual) in response to reviewer 3.
>> Included experiment 5 in the appendix for larger graphs (N = 6), with separation of equivalence classes, as well as drew attention to Experiment 4 in the appendix, to demonstrate generalizability, in response reviewers 2, 3, and 4.
>> Experiment 5 also compares the performance of the active agent with an agent with a random intervention policy as suggested by reviewer 4.
>> Added a section “Summary of results”.
>> Explained our specific choice of episode length and intervention values.
>> Clarified Footnote 1 in response to reviewer 3.
>> Included reward distribution plot in Appendix A in response to reviewer 2.

---

### Meta-Review · Area_Chair1 · 2018-12-20

**Confidence:** 5
**Recommendation:** Reject

**Metareview:**

The reviewers raised a number of concerns including insufficiently demonstrated benefits of the proposed methodology, lack of explanations, and the lack of thorough and convincing experimental evaluation. The authors’ rebuttal failed to alleviate these concerns fully. I agree with the main concerns raised and, although I also believe that the work can result eventually in a very interesting paper, I cannot suggest it at this stage for presentation at ICLR.